# Nutrition Literacy among Adolescents and Its Association with Eating Habits and BMI in Tripoli, Lebanon

**DOI:** 10.3390/diseases9020025

**Published:** 2021-03-29

**Authors:** Sara Taleb, Leila Itani

**Affiliations:** Department of Nutrition and Dietetics, Faculty of Health Sciences, Beirut Arab University, Riad El Solh, Beirut P.O. Box 11–5020, Lebanon; sarahtaleb494@gmail.com

**Keywords:** nutrition literacy, food habits, BMI

## Abstract

(1) Background: Adolescence is a period of increased autonomy and independent decision making; it determines health behaviors that can persist into the future. Individual factors like food choices and unhealthy lifestyle have an essential role in the development and prevention of obesity among adolescents and are associated with the nutrition literacy of parents and other adults. While the association of parents’ nutrition literacy with adolescent BMI has been addressed, there is still a scarcity of studies that examine the effect of adolescents’ nutrition literacy on their eating habits and body mass index (BMI) status. (2) Methods: A cross-sectional study was conducted that included 189 adolescents (68 with overweight and obesity and 121 with normal weight) aged between 14–19 years from four private schools in Tripoli, Lebanon. A self-administered questionnaire that included the Nutrition Literacy Assessment Instrument (NLAI) and the Adolescent Food Habits Checklist (AFHC) was used. Anthropometrics were measured using standardized procedures. The association between nutrition literacy, food habits and BMI was assessed using a chi squared test for independence and Poisson regression analysis where suitable. (3) Results: Results indicated no association between all five components of nutrition literacy and body mass index categories. Furthermore, there was no association between the Adolescent Food Habits Checklist and overweight or obese BMI status (RR = 0.947, 95%CI: 0.629–1.426) (*p* = 0.796). No association was observed between nutrition literacy and food habits, except for an inverse association with macronutrients literacy. (4) Conclusions: In conclusion, the study indicated that there was no association between the components of nutrition literacy with body mass index or with food habits, except for macronutrient literacy.

## 1. Introduction

Adolescence is a period characterized by increased autonomy and independent decision making and can determine health related behaviors, namely dietary intake [1,2]. During adolescence, nutrition-related behaviors develop as a result of changes in situations and responsibilities as molded by social pressure and social norms; these can influence nutrition knowledge and skills. Both healthy and unhealthy behaviors adopted during adolescence persist into adulthood [3,4,5]. Adolescents’ dietary patterns are characterized by skipping meals, frequent snacking, and consumption of fast food, processed foods and sugar-sweetened beverages [1,6]. These dietary patterns have health implications that persist throughout one’s life, like weight gain and increased risk of obesity, cardiovascular diseases, type-two diabetes, some types of cancers, sarcopenic obesity and impaired quality of life [6,7,8,9,10,11]. Nutrition-related problems and behaviors have been recently reported to be associated with nutrition literacy, especially among adolescents [12,13,14].

Nutrition literacy, defined as the ability to obtain, process, understand and use correct nutrition information and nutritional knowledge to make healthy food choices, is associated with overall healthy diets [15,16,17] and diet quality [18]. Moreover, nutrition literacy falls under the umbrella of health literacy; low health literacy in adolescents is associated with obesity [19] and lower levels of health-promoting behaviors [20,21,22,23]. Low nutrition knowledge and skills increase the susceptibility to excess weight gain and its associated health conditions [24].

In Lebanon, recent studies have suggested that a shift in adolescents’ dietary intake, characterized by adherence to a westernized pattern of sedentary lifestyle, is associated with overweight and obesity and its comorbidities [25,26,27,28,29,30], which can persist throughout one’s life and present health challenges [6,7,8,9,10,11]. With obesity rates far from global targets for 2025 (Global Nutrition report, 2025), adolescence provides an unequalled opportunity to establish healthy eating behaviors through health promoters, which could prevent obesity and related health problems later in life [31,32]. Understanding the determinants of healthy eating in this age group is essential in planning interventions at the public health level or in clinical settings to prevent comorbidities.

Studies in the literature have focused on health literacy and the effect of parent’s nutrition literacy on adolescents’ body mass index (BMI) rather than adolescents themselves [19,33]. However, up to the time of drafting this manuscript, data related to the influence of nutrition literacy on eating patterns and body weight was scarce, particularly in Lebanon. Hence, this study was designed to investigate the association between nutrition literacy among adolescents and their BMI status and food habits.

## 2. Materials and Methods

### 2.1. Study Participants

A sample of 210 adolescents (14–19 years) with a 1:2 ratio of adolescents with overweight (BMI for age Z score > +1 SD) or obesity (BMI for age Z score > +2 SD) or normal weight (−2 SD ≤BMI for age ≤ +1 SD) was used for this cross-sectional study. The sample size was estimated assuming 80% power and 95% confidence interval as well as a hypothetical 20% prevalence of low nutrition literacy among adolescents with normal weight and 40% among adolescents with overweight and obesity. Based on the sex distribution among adolescents with overweight and obesity in Lebanon [28], it was estimated that the study should include 70 adolescents with overweight or obesity (42 (60%) boys and 28 (40%) girls) and 140 (63 (45%) boys and 77 (55%) girls) adolescents with normal weight.

Participants were included based on the following criteria: (1) either overweight or normal weight, (2) aged between 14–19 years and (3) English educated. Students were excluded if (1) aged less than 14 or more than 19 years or (2) underweight, with a BMI < −2 SD for age Z score.

### 2.2. Data Collection

Data was collected from four private schools in Tripoli, North Lebanon. Schools were contacted after random selection from the national school guide (MEHE, 2015–2016). Four schools agreed to participate in the study and four classes (grades 9/10/11/12) were included in the study from each school. All participants in the same school and classroom were selected if they fulfilled the inclusion criteria and provided oral consent to participate. Quota sampling was used until the desired sample size was achieved. 

Body weight and height were measured using standardized techniques and calibrated equipment. Weight was measured to the nearest 0.1 kg, with participants wearing minimal indoor clothing, using a calibrated electronic scale (Amber Body scale APAFE002; NUMED SARL 2012, Beirut, Lebanon). Height was measured to the nearest 0.1 cm while the participant stood barefoot with feet together, arms at the sides and head in the Frankfort horizontal plane, using a portable stadiometer (Portable Height scale AHP003; NUMED SARL 2012, Beirut, Lebanon).

A self-administered questionnaire was used, consisting of three components. The first component retrieved information on the sociodemographic characteristics of the participants (age, sex, father employment status, mother employment status, mother education level, father education level, grade, monthly income of the family, place of residence, number of rooms in the house and number of individuals living in the house). The other two parts included two instruments: the Nutrition Literacy Assessment Instrument (NLAI) [24] and the Adolescent Food Habits Checklist (AFHC). The NLAI consisted of five subscales: Nutrition and health (NH), Macronutrients (MA), Household food measurement (HFM), Food label and numeracy (FLN), and Food group (FG). The two tools were scored as described elsewhere [24,34]. The total scores for the AFHC and NLAI subscales were categorized into tertiles. Adolescents scoring in the first tertile were given a score of 1, those scoring in the middle tertile were given a score of 2 and those scoring in the third tertile were given a score of 3. Adolescents were categorized dichotomously into either low or medium-high tertiles for nutrition literacy and food habits. The frequency of scoring in either the low tertile or medium-high tertile was calculated.

### 2.3. Data Analysis

Smirnov-Kolmogorov and Shapiro-Wilk tests were used to check the normality of the data. Means and standard deviations were calculated for continuous variables. A chi squared test was used to compare proportions. A student’s t-test was used to compare means. Means and standard deviations were calculated for continuous variables, or median IQR where appropriate, based on normality tests. The variability and reliability of the NLAI subscales were analyzed using box plots and Cronbach’s alpha, respectively (Appendix A). Frequencies and proportion were used for categorical variables. A Student’s t-test or Mann-Whitney test were used where appropriate to compare means or medians, respectively. A chi squared test for independence was used for proportions. Simple and multiple Poisson regression analysis was used to calculate the relative risk of being affected by overweight or obesity (BMI for age Z score > +1 SD), with inadequate nutrition literacy or its components as independent variables. Multiple Poisson regression was used while adjusting for potential confounders. All tests were considered statistically significant at *p* < 0.05. Statistical Package for the Social Sciences (SPSS) software [35] was used for all the analyses.

## 3. Results

### 3.1. Characteristics of the Study Population

A total of 189 participants completed the survey, aged between 14 and 19 years old and attending private schools, giving a response rate of (90%). Out of 189 individuals, 121 (64%) were with normal body weight and 68 (36%) demonstrated overweight and obesity.

Anthropometric characteristic differed significantly across BMI categories (Table 1). The median weight among adolescents with normal weight—56.00 kg (IQR: 52.00–62.30)—was lower compared to those with overweight and obesity—76.30 kg (IQR:68.35–90.00). BMI followed the same trend; it was lower for adolescents with normal weight—20.40 (IQR:19.00–21.89)—than for those with overweight and obesity—26.69 (IQR:25.25–29.00). Similarly, BMI for age Z score for adolescents with normal weight (0.016 ± 0.7) was significantly lower than that for those with overweight and obesity (1.8 ± 0.5). The participants did not differ significantly with respect to sociodemographic characteristics (Table 1).

### 3.2. The Level of Nutrition Literacy and Its Distribution in the Study Sample

Table 2 presents median (IQR) nutrition literacy scores and the distribution of adequate nutrition literacy in the study sample. Although the median nutrition literacy score for each subscale did not vary across the BMI categories, most of the study sample had adequate nutrition literacy with respect to NH (5.00 (IQR:4.00–6.00)) and MA (5.00 (IQR:4.00–5.00)) compared to a total score of six, as well as in FG (12.00 (IQR:11.00–14.00)) compared to a total score of 15. This exceeded 80% of the total possible score, indicating a greater than average level of nutrition literacy related to nutritional health (NH), macronutrients (MA) and food groups (FG). Nonetheless, the study sample exhibited marginal nutrition literacy for household food measure (HFM) (3.00 (IQR:2.00–3.00)) and food label reading (FLN) (3.00 (IQR:2.00–4.00)) compared to a total possible score of six.

### 3.3. Association between Nutrition Literacy and BMI Status

Table 3 presents the relative risk of having overweight or obesity with inadequate nutrition literacy in the different components of the nutrition literacy scale. The results indicated that there was no association between all the components of nutrition literacy (MA, NH, HFM, FLN and FG) and BMI categories (normal vs. overweight/obesity). This result held even after the adjustment for potential confounders (Table 3).

### 3.4. Healthy Eating Habit Scores and Distribution by BMI Status 

Table 4 presents the distribution of unhealthy eating habits among participants under study by BMI status. Overall, adolescents scored low on the AFHC (11.50 (IQR:8.00–15.00)) compared to the total score of 23 points. The mean AFHC score did not differ between adolescents with normal weight (11.50 (IQR:7.83–15.00)) or those with overweight and obesity (11.25 (IQR:8.36–14.45)). Overall, the majority of adolescents had a medium/high food habit score (64.6%), with the lowest being in adolescents with normal weight (63.6%), followed by adolescents with overweight and obesity (66.2%). However, there was insignificant difference in the distribution of AHFC items across categories of BMI (*p* > 0.05).

### 3.5. Association between Healthy Eating Score (AHFC) and BMI

When looking at the relative risk for having a low score on the AFHC, the result indicated that there was no association between the AFHC (RR = 0.947, 95% CI: 0.629–1.426) (*p* = 0.796) and BMI (normal weight vs. overweight and obesity) (Table 5). The results were adjusted for total score of the different nutrition literacy subscale to account for nutrition literacy.

### 3.6. Association between Healthy Eating Score and Nutrition Literacy

Table 6 presents the mean scores for the five components of the nutrition literacy scale by food habit category. Adolescents with a lower AHFC score had a higher MA literacy score (5.00 (IQR:4.00–5.00)) than those with a medium or high AHFC score (4.00 (IQR:3.00–5.00)), with a significant association (X^2^ = 10.57; *p*–value = 0.001) between higher nutrition literacy and lower food habit score. Although those scoring low on the food habits scale scored significantly higher on household food measures literacy (3.00 (IQR:2.00–3.00) vs. 2.00 (IQR:1.75–3.00)), the association was not significant (X^2^ = 0.009; *p*–value=0.924). On the other hand, the rest of the nutrition literacy components (NH, HFM, FLN and FG) did not vary between AHFC groups or have an association with the AFHC score (Table 6).

## 4. Discussion

The study evaluated nutrition literacy in a sample of Lebanese adolescents and its association with their food habits and BMI status. Participants had adequate nutrition literacy on all components except HFM and FLN. However, there was no association between all components of nutrition literacy and BMI or food habits of adolescents. The exception to this was macronutrient literacy, which was associated with unhealthy food habits among Lebanese adolescents.

The adequacy of nutrition literacy observed among study participants indicates their basic nutritional knowledge and their capacity to understand information about food items and food groups, read food labels and control their portions, and make informed decisions or advocate to others about food choices [15,36]. Although adolescents were nutritionally literate, there was no association between nutrition literacy components and eating habits, except for an inverse association with macronutrient literacy. The lack of association can be explained by the fact that literacy alone, in terms of factual food-related knowledge and health-related risks, can make an individual contemplate carrying out heathy eating behaviors [37]. Behavioral capabilities include functional and procedural knowledge, critical thinking, goal setting, and problem solving and self-regulation skills; these are a precondition for acting and gaining self-efficacy [38]. In addition, social determinism, in terms of modeling nutrition behaviors for teenagers to learn, is dependent on social environment, which consists of peers and parents; these influences at a young age can lead to long-term healthy eating behaviors [38,39,40,41]. Thus, unless nutrition literacy is coupled with essential behavior capabilities and accompanied by environmental support and collaborative action and partnership at multiple levels of influence, behavior change is unlikely [38,42,43]. Consistently, the current findings are in line with previous studies from different regions, including USA, Uganda, Egypt and Europe. These studies reported that adequate nutrition literacy among adolescents was not associated with adequate eating habits. In these studies, although adolescents demonstrated awareness of dietary guidelines or scored high on nutrition knowledge scales, they failed to choose low-fat lunches or desserts, healthy cafeteria items, or follow a Mediterranean diet in their everyday life [33,34,35,36,37,38,39,40,41,42,43,44,45,46,47,48,49,50].

The unhealthy eating habits observed among adolescents in this study are a common finding in this age group. In Lebanon, recent national studies have reported unhealthy food consumption patterns associated with overweight and obesity [26,27]. Unhealthy dietary patterns among adolescents are associated with adverse health implications, including weight gain and increased risk of obesity, cardiovascular diseases, type-two diabetes, hypertension, chronic kidney disease, some types of cancers [6,7,8,9] and sarcopenic obesity, in addition to impaired quality of life [51]. What makes this observation critical is that adolescence is a period of life during which eating behavior and food preferences are established, and these often track into adulthood [52,53,54], thus influencing adult health [54] and quality of life, both of which have an impact on healthcare resources [9,10,11]. Despite the lack of association between nutrition literacy, eating habits and weight status, recent literature has clearly attributed the increasing trend in obesity among adolescents in Lebanon to unhealthy food consumption patterns along with lower physical activity levels [26,51,55]. Hence, the current findings should not be underscored, as nutritional knowledge has an essential role in the development and prevention of obesity later in life [1,56]. Moreover, our findings were in agreement with previous studies on adolescents in Saudi Arabia and India [57]. On the other hand, Kalakan et al. (2016) reported significantly higher AFHC scores among adolescents with overweight compared to those with underweight and normal weight in Turkey [58].

The literature examining the effect of nutrition literacy on weight status is still scarce. However, the insignificant association between nutrition literacy and BMI is in line with previous reports from different countries, including USA [33], Egypt [47], countries in Europe [45,46], Turkey [58] and Australia [59]. 

To our knowledge, this is the first study in Lebanon to associate nutrition literacy with BMI and eating habits in a sample of Lebanese adolescents using validated tools. Nevertheless, the study holds few limitations regarding the representativeness of the sample of the Lebanese population, with a focus on private schools and higher socioeconomic status. In addition, parental nutrition literacy and physical activity were not measured. Higher socioeconomic status was previously reported in Lebanon to be associated with higher nutritional knowledge scores and energy and macronutrient intake compared to lower socioeconomic status among youth, irrespective of healthy eating [60]. Furthermore, the presence of high food insecurity was associated with lower adherence to a Mediterranean diet pattern among Lebanese adolescents [61]. In addition, the role of parental nutrition literacy in determining healthy eating was recently reported in the literature [14,62], which could have contributed to the observed nutrition literacy, despite the lower healthy eating score. Finally, although reliability analysis (Appendix A) of the NLAI subscales revealed small values for Cronbach’s alpha, this is justified by the small number of items (<10) that can have a profound effect on the alpha value [63,64].

## 5. Conclusions

Nutrition literacy in this age group, even if it is considered a necessary behavioral capability, is not sufficient to mold heathy eating behaviors or determine overweight and obesity. This confirms previously explained models for determinants of obesity and behavior change in children’s ecological niches [65,66], in addition to genetics, food preferences and increased autonomy, that interact with family, community, culture, and physical, social and political environments as well as food environments [2,43,66,67,68,69,70,71]. Therefore, appropriate levels of nutrition knowledge and literacy may not be enough to influence adolescents’ food habits or body weight. Hence, within the framework of a causal theory of obesity, early exposure to nutrition literacy can be studied further as a mediator of the interplay of the multiple influences that can, with time, increase autonomy and enhance eating behavior and diet quality, which line the causal pathway to adult obesity. Therefore, it is essential to unveil this role and enforce healthy eating by interventions delivered through a life course approach, specifically focusing on school-based food and nutrition education.

## Figures and Tables

**Table 1 diseases-09-00025-t001:** Sociodemographic and anthropometric characteristics of the study population (*n* = 189) ^ǂ^.

Variable	Overweight/Obese (*n* = 68)	Normal Weight (*n* = 121)	Total (*n* = 189)	Significance **
Age (years)	16 (14.25–17.00)	16.00 (15.00–16.00)	16.00 (15.00–17.00)	*p* = 0.396
Sex				X^2^ = 0.024, *p* = 0.878
Male	39 (57.4)	68 (56.2)	107 (56.6)	
Female	29 (42.6)	53 (43.8)	82 (43.4)	
Height (cm)	168.80 ± 9.94	166.17 ± 8.39	167.11 ± 9.04	*p* = 0.056
Weight (kg)	76.30 (68.35–90.00)	56.00 (52.00–62.30)	61.30 (54.60–71.00)	*p* < 0.0001
Body Mass Index (kg/m^2^)	26.69 (25.25–29.00)	20.40 (19.00–21.89)	22.00 (520.0–25.76)	*p* < 0.0001
BMI Z score	1.89 ± 0.57	0.016 ± 0.69	0.69 ± 1.11	*p* < 0.0001
Grade				X^2^ = 0.013, *p* = 0.908
Grade 9–10	32 (47.1)	58 (47.9)	90 (47.6)	
Grade 11–12	36 (52.9)	63 (52.1)	99 (52.4)	
School				X^2^ = 0.146, *p* = 0.702
High socioeconomic	50 (73.5)	92 (76.0)	142 (75.1)	
Low socioeconomic	18 (26.5)	29 (24.0)	47 (24.9)	
Mother education				X^2^ = 0.315, *p* = 0.854
Elementary and vocational school	18 (26.5)	32 (26.4)	50 (26.5)	
Secondary school	14 (20.6)	29 (24.0)	43 (22.8)	
University education	36 (52.9)	60 (49.6)	96 (50.8)	
Father education				X^2^ = 2.063, *p* = 0.357
Elementary and vocational school	24 (35.3)	46 (38.0)	70 (37.0)	
Secondary school	19 (27.9)	23 (19.0)	42 (22.2)	
University education	25 (36.8)	52 (43.0)	77 (40.7)	
Mother employment status				X^2^ = 2.222, *p* = 0.329
Employee	14 (20.6)	29 (24.0)	43 (22.8)	
Self-employed	8 (11.8)	7 (5.8)	15 (7.9)	
Unemployed	46 (67.6)	85 (70.2)	131 (69.3)	
Father employment status *				X^2^ = 2.465, *p* = 0.116
Employee	29 (42.6)	66 (54.5)	95(50.3)	
Self-employed	39 (57.4)	55 (45.5)	94(49.7)	
Monthly income of the family				X^2^ = 2.441, *p* = 0.118
<600,000 L.L.	19 (27.9)	22 (18.2)	41(21.7)	
>600,000 L.L.	49 (72.1)	99 (81.8)	148(78.3)	
Place of residence				X^2^ = 2.939, *p* = 0.086
Urban	55 (80.9)	84 (69.4)	139 (73.5)	
Rural	13 (19.1)	37 (30.6)	50 (26.5)	

^ǂ^ For continuous data, values are mean ± (SD) or Median (IQR), and for categorical data, values are n (%). * None of the fathers were unemployed. ** Significance pertains to chi squared or student *t* test or Mann-Whitney test between two BMI categories: normal weight and overweight/obesity.

**Table 2 diseases-09-00025-t002:** Median (IQR) of nutrition literacy scores and the distribution of adequate nutrition literacy by BMI category (*n* = 189) ^ǂ^.

Variable *^¥^	Overweight/Obese (*n* = 68)	Normal Weight (*n* = 121)	Total (*n* = 189)	Significance **
Total NH score	5.00 (4.00–6.00)	5.00 (4.00–6.00)	5.00 (4.00–6.00)	*p* = 0.868
				X^2^ = 0.837, *p* = 0.36
Inadequate and marginal nutrition literacy	13 (19.1)	17 (14.0)	30 (15.9)	
Adequate nutrition literacy	55 (80.9)	104 (86.0)	159 (84.1)	
Total MA score	4.50 (3.00–5.00)	5.00 (4.00–5.00)	5.00 (4.00–5.00)	*p* = 0.569
				X^2^ = 1.109, *p* = 0.292
Inadequate and marginal nutrition literacy	18 (26.5)	24 (19.8)	42 (22.2)	
Adequate nutrition literacy	50 (73.5)	97 (80.2)	147 (77.8)	
Total HFM score	2.00 (2.00–3.00)	3.00 (2.00–3.00)	3.00 (2.00–3.00)	*p* = 0.449
				X^2^ = 0.081, *p* = 0.776
Inadequate and marginal nutrition literacy	58 (85.3)	105 (86.8)	163 (86.2)	
Adequate nutrition literacy	10 (14.7)	16 (13.2)	26 (13.8)	
Total FLN score	3.00 (2.00–4.00)	3.00 (2.00–4.00)	3.00 (2.00–.00)	*p* = 0.938
				X^2^ = 0.014, *p* = 0.907
Inadequate and marginal nutrition literacy	41 (60.3)	74 (61.2)	115 (60.8)	
Adequate nutrition literacy	27 (39.7)	47 (38.8)	74 (39.2)	
Total FG score	12.00 (11.00–13.00)	12.00 (11.00–14.00)	12.00 (11.00–14.00)	*p* = 0.970
				X^2^ = 0.787, *p* = 0.375
Inadequate and marginal nutrition literacy	12 (17.6)	28 (23.1)	40 (21.2)	
Adequate nutrition literacy	56 (82.4)	93 (76.9)	149 (78.8)	

^¥^ NH: Nutrition and health; MA: Macronutrients; HFM: Household food measurement; FLN: Food label and numeracy; FG: Food group. ^ǂ^ For continuous data, values are median (IQR), and for categorical data, values are n (%). * For NH, MA, HFM and FLN: inadequate and marginal nutrition literacy (0 < score < 3); adequate nutrition literacy (4 < score < 6). For FG: inadequate and marginal nutrition literacy (0 < score < 10); adequate nutrition literacy (11 < score < 15) ** Significance pertains to chi squared or Mann-Whitney non-parametric test between 2 BMI categories: normal weight and overweight/obesity.

**Table 3 diseases-09-00025-t003:** The relative risk of being affected by overweight or obesity with inadequate nutrition literacy in the different components of the nutrition literacy scale (*n* = 189).

Nutrition Literacy Scale ^¥^	Nutrition Literacy Score	Model	
		Simple	Multiple *
		RR (95% CI)	
MA literacy	Adequate	1	1
	Inadequate and marginal	1.066 (0.946–1.201)	1.049 (0.929–1.185)
NH literacy	Adequate	1	1
	Inadequate and marginal	1.065 (0.93–1.219)	1.062 (0.931–1.21)
HFM literacy	Adequate	1	1
	Inadequate and marginal	0.979 (0.847–1.133)	0.937 (0.81–1.083)
FLN literacy	Adequate	1	1
	Inadequate and marginal	0.994 (0.897–1.102)	0.974 (0.877–1.081)
FG literacy	Adequate	1	1
	Inadequate and marginal	0.945 (0.836–1.069)	0.91 (0.807–1.026)

^¥^ NH: Nutrition and health; MA: Macronutrients; HFM: Household food measurement; FLN: Food label and numeracy; FG: Food group. * Model adjusted for age, sex, mother education, father education, mother employment, father employment, grade, school type and place of residence.

**Table 4 diseases-09-00025-t004:** Distribution of unhealthy eating habits among participants under study by BMI status (*n* = 189).

Variable	Overweight/Obese (*n* = 68)	Normal Weight (*n* = 121)	Total (*n* = 189)	Significance
Total AFHC score ^¥^	11.25 (8.36–14.45)	11.50 (7.83–15.00)	11.50 (8.00–15.00)	*p* = 0.842
AFHC categories				X^2^ = 0.123, *p* = 0.726
Low	23 (33.8)	44 (36.4)	67 (35.4)	
Medium and High	45 (66.2)	77 (63.6)	122 (64.6)	

^¥^ Values are median (IQR). AHFC: Adolescent Food Habits Checklist.

**Table 5 diseases-09-00025-t005:** Relative risk and 95% CI for having a low score on the Adolescent Food Habit Checklist (*n* = 189).

Variables	Model	
	Simple	Multiple *
	RR (95%CI)	
BMI Z score overweight/obesity		
Normal weight	1	1
Overweight and obesity	0.93 (0.619–1.398)	0.947 (0.629–1.426)

* Model adjusted for age, sex, grade, school, socioeconomic level, father employment status, mother employment status, mother education, father education, place of residence, total NH, total MA, total HFM and total FLN. NH: Nutrition and health; MA: Macronutrients; HFM: Household food measurement; FLN: Food label and numeracy; FG: Food group.

**Table 6 diseases-09-00025-t006:** Mean nutrition literacy score by food habits score category (*n* = 189) ^ǂ^.

Variable *	Low Food Habits Score (*n* = 67)	Medium High Food Habits Score (*n* = 122)	Total (*n* = 189)	Significance **
Total NH score	5.00 (4.00–6.00)	5.00 (4.00–6.00)	5.00 (4.00–6.00)	*p* = 0.328
				X^2^ = 2.288, *p* = 0.130
Inadequate and marginal nutrition literacy	7 (10.4)	23 (18.9)	30 (15.9)	
Adequate nutrition literacy	60 (89.6)	99 (81.1)	159 (84.1)	
Total MA score	5.00 (4.00–5.00)	4.00 (3.00–5.00)	5.00 (4.00–5.00)	*p* = 0.001
				X^2^ = 10.57, *p* = 0.001
Inadequate and marginal nutrition literacy	6 (9.0)	36 (29.5)	42(22.2)	
Adequate nutrition literacy	61 (91.0)	86 (70.5)	147 (77.8)	
Total HFM score	3.00 (2.00–3.00)	2.00 (1.75–3.00)	3.00 (2.00–3.00)	*p* = 0.042
				X^2^ = 0.009, *p* = 0.924
Inadequate and marginal nutrition literacy	58 (86.6)	105 (86.1)	163 (86.2)	
Adequate nutrition literacy	9 (13.4)	17 (13.9)	26 (13.8)	
Total FLN score	3.00 (2.00–4.00)	3.00 (2.00–4.00)	3.00 (2.00–4.00)	*p* = 0.126
				X^2^ = 1.739, *p* = 0.187
Inadequate and marginal nutrition literacy	45 (67.2)	70 (57.4)	115 (67.2)	
Adequate nutrition literacy	22 (32.8)	52 (42.6)	74 (39.2)	
Total FG score	12.00 (11.00–14.00)	12.00 (11.00–14.00)	12.00 (11.0–14.00)	*p* = 0.634
				X^2^ = 0.004, *p* = 0.947
Inadequate and marginal nutrition literacy	14 (20.9)	26 (21.3)	40 (21.2)	
Adequate nutrition literacy	53 (79.1)	96 (78.7)	149 (78.8)	

^ǂ^ For continuous data, values are median (IQR), and for categorical data, values are n (%). * For NH, MA, HFM and FLN: inadequate and marginal nutrition literacy (0 < score < 3); adequate nutrition literacy (4 < score < 6). For FG: inadequate and marginal nutrition literacy (0 < score < 10); adequate nutrition literacy (11 < score < 15) ** Significance pertains to chi squared or student t test between two categories of food habits scores. NH: Nutrition and health; MA: Macronutrients; HFM: Household food measurement; FLN: Food label and numeracy; FG: Food group.

## Data Availability

Are available from the corresponding author on reasonable request.

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
