# Peer review of "Nutrition Literacy among Adolescents and Its Association with Eating Habits and BMI in Tripoli, Lebanon"

_diseases, 2021, doi:10.3390/diseases9020025_

Round 1
Reviewer 1 Report
Report on “Nutrition Literacy among Adolescents and its Association with Eating Habits and BMI in Tripoli-Lebanon” by Sara Taleb and Leila Itani
The manuscript investigates to investigate the association between nutrition literacy among adolescents and their BMI status and food habits. I think the manuscript it is well written and quite clear. My concerns lie in the lack of variability in the results of the questionnaires and some points of the methodology:
Concerns:
- I am not sure about if NH score, MA score, HFM score, FLN score, and FG score have enough variability in their results to provide significative differences in the study. The reliability of the questionnaires should be analyzed. You can calculate the alpha Cronbach values for each questionnaire or even show variability in their results using boxplots for each score. It needs a more detailed description of the results of the scores. In addition, this must be explained as a limitation of the study in the discussion section.
- Data Analysis. The authors declare expressed continuous variables by mean and standard deviation. This is not advisable, only in case that these characteristics follow a normal distribution can be reported in that way. I don’t think that weight or BMI follows a normal distribution in every group. General characteristics can be reported using median and interquartile range, or in the first term, you must test if variables follow a normal distribution using Kolmogorov-Smirnov or Shapiro Wilk test, and then variables can be reported as mean and standard deviation.
- Data Analysis. The authors used Independent t-test to determine whether there was a significant difference in groups based. To do this, normal distribution of variables must be checked, if not, Wilcoxon Mann-Whitney test must be used for this purpose.
- Material and Methods. Study participants. The weight groups are wrong described: Change over-weight or obesity (BMI for age Z score > +1 SD) or normal weight (2 SD ≤ BMI for age ≤ +1 SD) by over-weight (BMI for age Z score > +1 SD) or obesity (BMI for age Z score > +2 SD) or normal weight (BMI for age ≤ +1 SD)
Author Response
We thank the reviewer for the thorough and constructive review

Reviewer 2 Report
The topic of the study is interesting and important given the global high rates of obesity, which are alarming especially among the youth.
I have a couple of comments:
- On rows79-80. Same criteria given twice. Maybe delete exclusion #1.
- Table 3. All abbreviations should be written out (e.g. in text or in the table footnote). Now it is impossible to understand what the types of nutrition literacy consists of.
You write about macronutrient literacy, micronutrient literacy etc but no explanation how exactly these are measured.
Overall this is well written.
Author Response
We thank the reviewer for the thorough revision and constructive comments which enhanced greatly the manuscript.
Reviewer 2 :
The topic of the study is interesting and important given the global high rates of obesity, which are alarming especially among the youth.
I have a couple of comments:
- On rows 79-80. Same criteria given twice. Maybe delete exclusion #1
Response: Done as suggested
- Table 3. All abbreviations should be written out (e.g. in text or in the table footnote). Now it is impossible to understand what the types of nutrition literacy consists of.
Response: Abbreviations added as indicated
- You write about macronutrient literacy, micronutrient literacy etc but no explanation how exactly these are measured. Overall this is well written.
Response: These are measured as subscales of the NLAI. The NLAI consisted of five subscales namely, Nutrition and health (NH), Macronutrients (MA), Household food measurement (HFM), Food label and numeracy (FLN), Food group (FG).
Micronutrient literacy was used unintentionally as a typo mistake and corrected in the manuscript

Reviewer 3 Report
This is an interesting study looking at the associations between nutrition literacy, self-reported eating habits and body weight among normal weight and overweight adolescents attending four private schools in Lebanon. The authors reported finding no significant associations between nutrition literacy, eating habits and body weight, suggesting that adequate nutrition literacy alone does not necessarily predict healthy eating behaviors. Perhaps, paradoxically, the authors also reported that higher nutrition literacy was associated with lower food habits score, although this was not related to body weight status.
The paper is generally well written although could be improved as suggested below:
Methods section: The authors should define the various subscales for the NLAI assessment, as they begin using acronyms for the subscales starting on line 141 and later in the paper (e.g., line 155). These subscales should be spelled out the first time they are used, giving the acronym abbreviation.
The reference given for the NLAI (24) is a graduate thesis reporting validation done in adults. Is there a reference for this instrument being validated for adolescents of the age range used in this report? E.g., The Adaptation of the Adolescent Food Habit Checklist to the Turkish Adolescents, January 2012TAF preventive medicine bullletin 11(1):45, DOI: 10.5455/pmb.20110914125525
Discussion: There really isn’t much discussion of the findings other than to say they are consistent with studies in other countries. The findings seem to underscore the frequent observation that nutrition education and literacy are not sufficient to motivate healthy eating behavior. What do the authors make of the finding that higher nutrition literacy was associated with a lower food habits score? This seems counter intuitive.
The authors also do not suggest other possible explanations for the reason some adolescents may be overweight despite adequate nutrition literacy (e.g., poor physical activity habits?).
The authors point out limitations (population representation, parents’ nutrition literacy), although they do not discuss the impact these limitations may have had on their results and conclusions. For example, the adolescent sample was selected from private schools, which presumably would represent a higher socio-economic status (SES) compared to the general population (at least this would be true in some countries like the USA). Although they provided demographic information about the adolescents and parents, they did not compare that to the general population in Lebanon. And, how does the monthly income dichotomy presented in the demographic table represent the income distribution of the general population? In other words, might their findings only apply to adolescents coming from mid to upper income households having similar SES as those in their study? What might be known about other segments of the population and are there hypothesized differences (e.g., what would they expect among adolescents with poor nutrition literacy?).
Finally, the authors propose the importance of developing healthy eating habits early in life to prevent deleterious effects into adulthood. What did they learn from this study that would inform future studies and interventions on promoting healthy eating? … because they showed that nutrition literacy may not be sufficient (even if necessary).
Other comments:
There are some typographical errors (presumably) that may confuse readers. For example, on line 28 of the abstract the authors describe “micronutrient” literacy as a characteristic related to food habits. This is presumably supposed to be “macronutrient” literacy, which is the subscale measured by the NLAI assessment instrument. This error occurs a few more times throughout the manuscript, and is important to correct for readers.
Line 200; change the words “effects on” to “associations with”. This is an observational study and cannot determine cause and effect.
Line 209; eliminate the word “yet”
Author Response
We thank the reviewer for the thorough revision and constructive comments which enhanced greatly the manuscript.
Reviewer 3:
This is an interesting study looking at the associations between nutrition literacy, self-reported eating habits and body weight among normal weight and overweight adolescents attending four private schools in Lebanon. The authors reported finding no significant associations between nutrition literacy, eating habits and body weight, suggesting that adequate nutrition literacy alone does not necessarily predict healthy eating behaviors. Perhaps, paradoxically, the authors also reported that higher nutrition literacy was associated with lower food habits score, although this was not related to body weight status.
The paper is generally well written although could be improved as suggested below:
- Methods section: The authors should define the various subscales for the NLAI assessment, as they begin using acronyms for the subscales starting on line 141 and later in the paper (e.g., line 155). These subscales should be spelled out the first time they are used, giving the acronym abbreviation.
Response: Done as suggested
- The reference given for the NLAI (24) is a graduate thesis reporting validation done in adults. Is there a reference for this instrument being validated for adolescents of the age range used in this report? E.g., The Adaptation of the Adolescent Food Habit Checklist to the Turkish Adolescents, January 2012TAF preventive medicine bullletin 11(1):45, DOI: 10.5455/pmb.20110914125525
Response: Nutrition literacy is a recent concept and up to our knowledge at the time of data collection there were no age specific tool.
- Discussion: There really isn’t much discussion of the findings other than to say they are consistent with studies in other countries. The findings seem to underscore the frequent observation that nutrition education and literacy are not sufficient to motivate healthy eating behavior. What do the authors make of the finding that higher nutrition literacy was associated with a lower food habits score? This seems counter intuitive.
Response: an explanation on the lack of association between nutrition knowledge or literacy and eating behavior was added the discussion.
- The authors also do not suggest other possible explanations for the reason some adolescents may be overweight despite adequate nutrition literacy (e.g., poor physical activity habits?).
Response: done as suggested
- The authors point out limitations (population representation, parents’ nutrition literacy), although they do not discuss the impact these limitations may have had on their results and conclusions. For example, the adolescent sample was selected from private schools, which presumably would represent a higher socio-economic status (SES) compared to the general population (at least this would be true in some countries like the USA). Although they provided demographic information about the adolescents and parents, they did not compare that to the general population in Lebanon. And, how does the monthly income dichotomy presented in the demographic table represent the income distribution of the general population? In other words, might their findings only apply to adolescents coming from mid to upper income households having similar SES as those in their study? What might be known about other segments of the population and are there hypothesized differences (e.g., what would they expect among adolescents with poor nutrition literacy?).
Response:
Regarding selection of private schools, this was limited by the restriction from the Ministry of education and higher education (MEHE) on access to public schools by researchers. However, According the central administration of statistics “In 2009-2010 there were 2,882 schools in total in Lebanon with the majority of pupils were enrolled in private schools, accounting for 66% of pupils. Which shifted in 2018-2019 to account between 45 and 55 % of pupils 10-19 years old being enrolled in private schools.
The dichotomy of wages was based on minimum wage level in Lebanon set in 2012 at 675000L.L from 500,000 L.L. in 2011(Minimuwage.org, 2021). According to CAS low paid worker <633,000 L.L. constituted 27.9 % of north residents and 21.8 % of total Lebanese.
Minimuwage.org : https://www.minimum-wage.org/international/lebanon#:~:text=What%20is%20the%20Lebanon%20Minimum,pounds%20(%24450)%20per%20month
International labor organization: https://www.ilo.org/dyn/travail/travmain.sectionReport1?p_lang=en&p_structure=1&p_year=2011&p_start=1&p_increment=10&p_sc_id=1&p_countries=LB&p_countries=MA&p_print=Y
Ref CAS: Central administration of statistics2019-2018. Labor Force and Household Living Conditions Survey (LFHLCS) 2018-2019 Lebanon. http://www.cas.gov.lb/index.php/demographic-and-social-en/education-en
- Finally, the authors propose the importance of developing healthy eating habits early in life to prevent deleterious effects into adulthood. What did they learn from this study that would inform future studies and interventions on promoting healthy eating? … because they showed that nutrition literacy may not be sufficient (even if necessary).
Response: done as suggested.
Other comments:
- There are some typographical errors (presumably) that may confuse readers. For example, on line 28 of the abstract the authors describe “micronutrient” literacy as a characteristic related to food habits. This is presumably supposed to be “macronutrient” literacy, which is the subscale measured by the NLAI assessment instrument. This error occurs a few more times throughout the manuscript, and is important to correct for readers.
Response: We thank the reviewer, actually it is a typo mistake, corrected as suggested.
- Line 200; change the words “effects on” to “associations with”. This is an observational study and cannot determine cause and effect.
Response: Done as suggested
- Line 209; eliminate the word “yet”
Response: Done as suggested

Round 2
Reviewer 1 Report
Most of my concerns have been raised by authors, I appreciate the effort that the authors have made in order to improve the manuscript.
There are only two minor concerns, first of them is that authors must include in the methods section a reference to the supplementary material where they included the variability analysis, and the second one is that authors must comment as a limitation of the study, in the discussion section, their results in the reliability analysis.
Best regards
Author Response
We thank the reviewer for all the constructive comments.
Reviewer1: Most of my concerns have been raised by authors, I appreciate the effort that the authors have made in order to improve the manuscript.
There are only two minor concerns, first of them is that authors must include in the methods section a reference to the supplementary material where they included the variability analysis, and the second one is that authors must comment as a limitation of the study, in the discussion section, their results in the reliability analysis.
Best regards
Response:
done as requested